# Comparative Safety Profiles of Oncology Biosimilars vs. Originators in Europe: An Analysis of the EudraVigilance Database

**DOI:** 10.3390/cancers15143680

**Published:** 2023-07-19

**Authors:** Victoria Nikitina, Greta Santi Laurini, Nicola Montanaro, Domenico Motola

**Affiliations:** 1Unit of Pharmacology, Department of Medical and Surgical Sciences, Alma Mater Studiorum, University of Bologna, Via Irnerio 48, 40126 Bologna, Italy; victoria.nikitina2@unibo.it (V.N.); greta.santilaurini2@unibo.it (G.S.L.); 2Alma Mater Studiorum, University of Bologna, 40126 Bologna, Italy; nicola.montanaro@unibo.it

**Keywords:** bevacizumab, biosimilars, monoclonal antibodies, oncologic biosimilars, oncology, pharmacovigilance, rituximab, trastuzumab

## Abstract

**Simple Summary:**

Nowadays, biosimilar drugs are numerous and widely used in many clinical fields, including oncology. However, skepticism remains towards these products among doctors and patients, particularly regarding their safety profile compared to the reference products. This prompted this comparative pharmacovigilance study using real-world clinical data. Consistent with the expected similarity in safety, our results reaffirm that biosimilars are comparable to the reference products in the real-world setting. This should further reassure and encourage their even greater use which, on the one hand, allows for all patients to be treated with the best available treatments and, on the other, frees up healthcare resources for innovative and more expensive drugs.

**Abstract:**

In the last decades, the clinical management of oncology patients has been transformed by the introduction of biologics. The high costs associated with the development and production of biologics limit patient access to these therapies. The expiration of exclusive patents for biologics has led to the development and market introduction of biosimilars, offering the reduction of costs for cancer treatments. Biosimilars are highly similar to the reference products in terms of structure, biological activity, efficacy, safety, and immunogenicity. Therefore, the monitoring of biosimilars’ safety in real-world clinical practice though pharmacovigilance is essential. This study aimed to analyze the post-marketing pharmacovigilance data of biosimilar monoclonal antibodies used in oncology and compare them with respective reference products. Data of a 2-year period (1 January 2021–31 December 2022) were retrieved from EudraVigilance, and descriptive and comparative analysis were performed using the Reporting Odds Ratio to evaluate the distribution of medicine-reaction pairs related to biosimilars of three antitumor biological products and their corresponding reference products: bevacizumab, rituximab, and trastuzumab. The results showed that most frequently reported ADRs for biosimilars were non-serious and consistent with the safety profiles of reference products. These findings provide reassurance regarding safety equivalence of biosimilars and support their use as valid alternatives to originator biologics.

## 1. Introduction

In the last decades, the clinical management of oncologic patients has undergone substantial changes due to the introduction of biologic medicines, which may contribute to a positive clinical outcome. Since biologics presents a complex structure and are produced in living system under strictly controlled conditions, their development and manufacture require very high costs, and consequently, these high prices are hardly affordable by the health service structures. Following the expiry of the exclusive patents for these biologic medicines, several biopharmaceutical industries have developed and introduced into the market similar biologic products, named biosimilars, allowing for a significant price lowering with a cut costs for cancer treatments.

Biosimilar is defined by the European Medicines Agency (EMA) as “a biological medicine highly similar to another already approved biological medicine (the ‘reference medicine’). Biosimilars are approved according to the same standards of pharmaceutical quality, safety and efficacy that apply to all biological medicines” [1]. Other definitions of biosimilars according to major international regulatory organizations are reported in Table 1. The EMA also specifies that biosimilar agents are highly similar to the reference product in terms of structure, biological activity and efficacy, safety, and immunogenicity profile [1]. EMA like other regulatory agencies has delineated specific comparability pathways to ascertain the similarity between a biosimilar candidate and its reference product (RP) [2,3,4]. Biosimilars differ from the generic form of the chemical products, because in this case it is not possible to develop molecules identical to their reference products due to biologic natural variation and their heterogenic product process [5]. Therefore, it is fundamental that the safety profiles of biosimilars are similar to those of the original biologics. Rituximab, bevacizumab, and trastuzumab are monoclonal antibodies (mAbs) used in first- and second-line treatment regimens in combination with other anti-cancer therapies for a number of common malignant diseases [6,7,8]. A growing number of biosimilar versions of these agents are available on the market, and many others are in development. In these circumstances, as with any medication, the evaluation of the adverse drug reaction (ADR) profiles of biologics and biosimilars through post-marketing surveillance is essential.

The aim of this study was to analyze the post-marketing pharmacovigilance data of biosimilar mAbs used in oncology and compare their safety information with the respective reference products.

## 2. Materials and Methods

Data were retrieved from the European Union’s post-marketing surveillance database EudraVigilance (EV), using the online interface adrreports.eu [11]. EV is a public spontaneous reporting system maintained by the European Medicines Agency on behalf of the European Union (EU), which receives Individual Case Safety Reports (ICSRs) of suspected adverse drug reactions within the European Economic Area (EEA) [12].

We retrieved all reports related to biosimilars of three antitumor biological products reported as suspected drugs bevacizumab, rituximab, and trastuzumab, and we compared their safety profile to the corresponding reference products (*Avastin*^®^, *MabThera*^®^, and *Herceptin*^®^, respectively). We considered the EU-licenced biosimilars which have been authorized before the year 2021: three different biosimilar bevacizumabs (*Aybintio*^®^, *Mvasi*^®^, *Zirabev*^®^), five biosimilar rituximabs (*Blitzima*^®^, *Rixathon*^®^, *Riximyo*^®^, *Ruxience*^®^, *Truxima*^®^), and six biosimilar trastuzumabs (*Herzuma*^®^, *Kanjinti*^®^, *Ogivri*^®^, *Ontruzant*^®^, *Trazimera*^®^, *Zercepac*^®^). Table 2 lists these medicine products with their respective dates of approval and the patent expiry dates for the corresponding reference products.

We performed our retrospective analysis on biosimilars, which have more time on the market, and take into account that two of them received a marketing authorization valid throughout the European Union in 2020 (*Ruxience*^®^ on 1 April 2020 and *Aybintio*^®^ on 19 August 2020); we considered the 2-year period between 1 January 2021 and 31 December 2022 in order to analyze and compare as many biosimilars as possible over an equal period of time.

### 2.1. Descriptive Analysis

The extracted reports were identified by a unique EU Local Number, reporting information on the report type (spontaneous or from clinical studies), primary source qualification (healthcare professional or non-healthcare professional), EV gateway receipt date, patient sex and age group, MedDRA preferred terms (PT), seriousness criteria, and suspect and concomitant drugs. All ADRs are categorized using the Medical Dictionary for Regulatory Activities (MedDRA), a specific standardized medical terminology to facilitate sharing of regulatory information internationally for medical products used by humans. MedDRA terms are arranged in a five-tiered multi-axial hierarchy, which provides increasing specificity as one descends it [13]. At the top-level of hierarchy there are 27 System Organ Classes (SOC), which incorporate at the lower levels High Level Group Terms (HLGTs) and High Level Terms (HLTs). Each member of the next level, Preferred Terms (PTs), is a distinct descriptor (single medical concept) for a symptom, sign, disease diagnosis, therapeutic indication, investigation, surgical or medical procedure, and medical social or family history characteristic. Finally, one or more Lower Level Terms (LLTs) correspond to each PT; the LLTs are effectively entry terms that include synonyms and lexical variants [14,15,16]. One or more symptoms can be reported for each EV report. We analyzed all the reports related to the reference drugs and their biosimilars included in this study performing a descriptive analysis. For each drug, the notoriety of the adverse reactions was ascertained by checking if the most frequently reported ADRs were listed in the corresponding Summary of the Product Characteristics (SPCs) [17,18,19,20,21,22,23,24,25,26,27,28,29,30,31,32,33].

### 2.2. Statistical Analysis

We performed comparative analysis using the Reporting Odds Ratio (ROR) with 95% confidence interval as statistical parameter to evaluate medicine-reaction pairs distribution. ROR allows for a quantitative approach using 2 × 2 contingency tables, comparing the frequency of a drug-reaction pair with all the others in the database. An increased frequency for the drug-reaction pair can be assumed if ROR is >1. The biosimilars and their respective original products were analyzed separately. A disproportionality analysis was carried out between the ADRs associated to biosimilars and each respective reference product. The EMA provides guidance on signal detection and management in pharmacovigilance to define a signal of disproportionate reporting in the EV system, and these criteria were used in this present study. The following criteria were applied: the lower bound of the 95% confidence interval greater than one; the number of individual cases greater than or equal to three for active substances contained in medicinal products included in the additional monitoring list in accordance with REG Art 23 (see GVP Module X), unless the sole reason for inclusion on the list is the request of a post-authorization safety study (PASS); five for the other active substances; and the event belongs to the Important Medical Event list [34]. (https://www.ema.europa.eu/en/documents/other/screening-adverse-reactions-eudravigilance_en.pdf. Accessed on 31 March 2023)

## 3. Results

### 3.1. Descriptive Analysis

Figure 1 describes the number of reports for bevacizumab, trastuzumab, and rituximab biologics and their respective biosimilars. Table 3, Table 4 and Table 5 summarize the characteristics of each report by patients’ sex (female, male, and unknown), age range (0–17 years, 18–64 years, 65–85 years, over 85 years, and unknown), and type of reporter (healthcare professional, and non-healthcare professional) for all medicine products in study. A total of 13,306 reports were collected for these products: 9806 reports (74%) referred to the original products and 3500 (26%) to biosimilars.

We analyzed a total of 36,200 reported PTs; 23,592 (65%) PTs were identified from reports for original products and 12,608 (35%) PTs for biosimilars. The proportions of reports from females were higher than from males for both categories: 58.5% vs. 36.2% for original products, and 57.6% vs. 26% for biosimilars. However, in 16.4% reports for the biosimilars, the information on sex was missing. Most of the reports referred to patients aged 18–64 years and a slightly lower percentage to those of the age group 65–85 years. Roughly 20% of the reports concerning originator biologics and biosimilars were missing information about patients’ age. Approximately 90% of reports were submitted by healthcare professionals.

### 3.2. Statistical Analysis

For the disproportionality analysis, all 13,306 safety reports had been examined, corresponding to 36,200 drug-reaction pairs. Reported ADRs referring to incorrect product storage, routine laboratory tests or incorrect administration, were not considered because they were not pertinent to our investigation. Overall, almost all of the most frequently reported and statistically significant ADRs for biosimilars were non-serious and listed in the corresponding SPCs. Bevacizumab biosimilars vs. *Avastin*^®^: *asthenia n* = 32 reactions, ROR 2.63 [CI 95% 1.81–3.82], *neuropathy peripheral n* = 29, ROR 2.44 [CI 95% 1.65–3.61], *abdominal pain n* = 28 ROR 3.92 [CI 95% 2.58–5.94]. Rituximab biosimilars vs. *MabThera*^®^: *COVID-19 n* = 169 reactions, ROR 1.73 [CI 95% 1.42–2.09], *pruritus n* = 163, ROR 1.96 [CI 95% 1.6–2.41], *throat irritation n* = 135 ROR 4.9 [CI 95% 3.55–6.74]. Trastuzumab biosimilars vs. *Herceptin*^®^: *diarrhea n* = 78 reactions, ROR 1.55 [CI 95% 1.23–1.95], *chills n* = 68, ROR 2.07 [CI 95% 1.61–2.66], *nausea n* = 57 ROR 1.6 [CI 95% 1.22–2.1]. Table 6, Table 7 and Table 8 show the most frequently reported and statistically significant ADRs for biosimilars compared to their original products by number of ADRs. Among these drug-reaction pairs we identified (on the basis of the Important Medical Event terms list [34]), serious reactions are listed below. Bevacizumab biosimilars vs. *Avastin*^ ®^: *neuropathy peripheral n* = 29 reactions, ROR 2.44 [CI 95% 1.65–3.61], *neutropenia n* = 24 reactions, ROR 1.75 [CI 95% 1.14–2.67], *thrombocytopenia n =* 18 ROR 2 [CI 95% 1.22–3.3], *seizure n =* 13 ROR 4.42 [CI 95% 2.3–8.5]. Rituximab biosimilars vs. *MabThera*^ ®^: *rheumatoid arthritis n =* 69 reactions, ROR 4.62 [CI 95% 2.76–7.74], *immune thrombocytopenia n =* 59 reactions, ROR 9.22 [CI 95% 3.71–22.88], *mantle cell lymphoma n =* 34 ROR 15.9 [CI 95% 1.6–158.05], *systemic lupus erythematosus n =* 34 ROR 7.95 [CI 95% 2.13–29.65]. Trastuzumab biosimilars vs. *Herceptin*^ ®^: *neutropenia n =* 32 reactions, ROR 3.29 [CI 95% 2.23–4.87], *thrombocytopenia n =* 16 ROR 2.59 [CI 95% 1.48–4.53], *hypokalaemia n =* 7 reactions, ROR 2.52 [CI 95% 1.03–6.16]. Furthermore, we analyzed the drug-reaction pairs with a higher and statistically significant ROR, which are listed as follows. Bevacizumab biosimilars vs. *Avastin*^®^: *weight increased n =* 14 reactions, ROR 8.58 [CI 95% 4.17–17.65], *general physical deterioration n =* 12 reactions, ROR 8.16 [CI 95% 3.72–17.9], *chills n =* 12 ROR 5.65 [CI 95% 2.76–11.58]. Rituximab biosimilars vs. *MabThera*^®^: *blood pressure fluctuation n =* 116 reactions, ROR 27.36 [CI 95% 8.55–87.51], *heart rate irregular n =* 48 reactions, ROR 22.48 [CI 95% 2.38–211.89], *ear pruritus n =* 39 ROR 18.24 [CI 95% 1.88–177.4]. Trastuzumab biosimilars vs. *Herceptin*^®^: *hypertransaminasaemia n =* 8 reactions, ROR 18.77 [CI 95% 3.52–100.17], *bronchospasm n =* 4 ROR 9.36 [CI 95% 1.32–66.58], *paraesthesia n =* 34, ROR 7.35 [CI 95% 4.8–11.26].

## 4. Discussion

Biosimilars require the submission of a Risk Management Plan, including further safety study, for their approval to be granted. Their safety profile is also monitored through pharmacovigilance activities once they are on the market, in the same way as the other medicines [1,3]. The spontaneous reporting system remains a cornerstone of pharmacovigilance since it allows the early detection of possible safety signals and the continuous monitoring and evaluation of potential safety issues in relation to reported ADRs [35]. In particular, it is crucial to identify rare or long-term ADRs that may not have been captured during the limited duration and controlled settings of premarketing clinical trials [36,37,38]. Spontaneous reporting system represents an important tool for identifying previously unknown ADRs and emerging safety issues since the data are collected from a wide range of healthcare providers, including physicians, pharmacists, and also patients. Moreover, post marketing data encompass a broader patient population compared to those involved in strictly controlled clinical trials settings, including, for instance, those with co-morbidities, concomitant medications, and vulnerable groups. This allows the system to reflect data on general population, real-world clinical practice, and patient experiences. However, pharmacovigilance studies such as this present one, based on spontaneous reporting system, have some limitations. First, the lack of denominator data, such as the number of patients exposed to a particular drug, does not allow for the accurate calculation of incidence rates, or determining the true risk associated with the use of a specific medication. Secondly, the information contained in the reports may be incomplete and inaccurate, may lack essential details, have inconsistencies, or be subjected to data entry errors. The quality and completeness of the reported data can affect the reliability and validity of the data. In addition, the lack of comprehensive information on the patient’s medical history, the usage of concomitant medications, and other relevant factors make it difficult to assess a causal relationship between the medication and reported ADR. Finally, and above all, the system is affected by the drawback of underreporting, owing to various factors such as lack of awareness, uncertainty about causality, lack of time or perception that reporting is burdensome [39,40,41,42]. This last limitation can affect the validity and accuracy of ROR estimates since its calculations are based on the assumption that the reporting rate is constant across all medications and ADRs. Furthermore, Reporting Odds Ratio does not intend to establish a causality relation between a drug and a given adverse reaction but simply to detect a safety signal. To overcome these limitations, it is crucial to integrate ROR with other pharmacovigilance methods, such as signal detection algorithms, disproportionality analyses, and additional observational or experimental studies. These complementary approaches can provide a more comprehensive understanding of the safety profile of a drug and minimize the impact of limitations.

This study provided post-marketing pharmacovigilance evidence in ADR reporting of originator biologics and corresponding biosimilars marketed in the EU, focusing on reported ADRs and the detection of disproportionality. Overall, during the study period the number of reports in EV for reference products has slightly increased in 2022 compared to the previous year, meanwhile, the number of reports for biosimilars has noticeably raised. This trend is evident for biosimilars of rituximab, most of reports referred to *Ruxience*^®^ and *Truxima*^®^, even though the number of reports for the reference product *MabThera*^®^ was a bit lower in 2022. The increasing number of reports for biosimilars in EV may reflect the significant increase in biosimilar usage in the EU in response to incentive programs instituted by individual member states, health authorities, and payers over the last few years [43,44]. In this contest, it is important to mention that the EU is leading the way with regard to the approvals, utilization, and realization of cost savings of biosimilars [45,46].

Almost all of the most frequently reported ADRs for analyzed biologics were non-serious, listed in the corresponding SPCs and in line with the studies by Giezen et al. [47,48,49]; thus, the adverse reactions to biologics that are included in their safety profiles are mostly related to their pharmacologic actions and immunologic reactions, e.g., immunogenicity and injection site reactions. Among serious, not listed in SPCs, and statistically significant ADRs, we focused on three adverse reactions of rituximab biosimilars: *rheumatoid arthritis*, *mantle cell lymphoma,* and *systemic lupus erythematosus*. Specifically, the signal of disproportionate reporting for *rheumatoid arthritis* referred to *Ruxience*^®^ and *Truxima*^®^, while *mantle cell lymphoma* and *systemic lupus erythematosus* only to *Truxima*^®^. We observed that in almost all reports that had those three ADRs, *Ruxience*^®^, and *Truxima*^®^ were used for rheumatoid arthritis, while *Truxima*^®^ for mantle cell lymphoma and systemic lupus erythematosus. In particular, *Truxima*^®^ was used off-label for the last two indications. These three reported ADRs lead to different interpretations. It can be argued that it is simply a writing mistake of the reporters, with *rheumatoid arthritis*, *mantle cell lymphoma*, and *systemic lupus erythematosus* being the target disease for the molecules in study rather than an adverse reaction. From another side, it cannot be excluded that the mention of the target disease in the field of adverse reactions may be understood as a statement of the disease progression due to therapeutic ineffectiveness. The high number of these PTs and their statistically significant ROR in our data makes this second interpretation preferable. These may be investigated by future pharmacoepidemiology studies. However, it is essential to acknowledge that ADR reporting can be influenced by several factors, such as drug usage patterns, and the underreporting is an important limit of spontaneous reporting databases like EV.

The results of this study show an overall comparability in safety profiles between the biosimilars and original biologics. Our findings are also in line with those of Kurki et al. [50], who investigated data on post-marketing safety of biosimilar mAbs up to 7 years post-approval, and observed no significant differences between the safety profiles of biosimilars and their reference products, no new or unexpected adverse reactions and no differences in the ADRs’ severity. The evidence acquired over 10 years of clinical experience shows that biosimilars approved through EMA can be used as safely and effectively in all their approved indications as other biological medicines [1]. Biosimilars have the potential to create a more sustainable healthcare system since the number of biosimilars in development in oncology is rising, especially biosimilars of bevacizumab, rituximab, and trastuzumab [51]. Copies of original biologics and biosimilars have lowered the costs and increased access to biologicals throughout the EU. Savings from the impact of biosimilar competition continues to grow: as of 2022, the cumulative savings at list prices from the impact of biosimilar competition in Europe reached over EUR 30 billion [52]. Therefore, biosimilars are key in significant economic savings of healthcare systems, allowing more patients to access modern therapies while offering comparable safety and efficacy profiles.

## 5. Conclusions

Biological therapies are a cost-effective alternative that have revolutionized the treatment of oncologic diseases. Based on the analysis of ADR reports from EudraVigilance, there were no significant differences in the safety profiles between bevacizumab, trastuzumab, and rituximab biosimilars and their respective originators in Europe. These findings provide reassurance regarding the safety equivalence of biosimilars and support their use as viable alternatives to originator biologics. As with any medication, constant pharmacovigilance monitoring is essential to ensure the ongoing safety of these products.

## Figures and Tables

**Figure 1 cancers-15-03680-f001:**
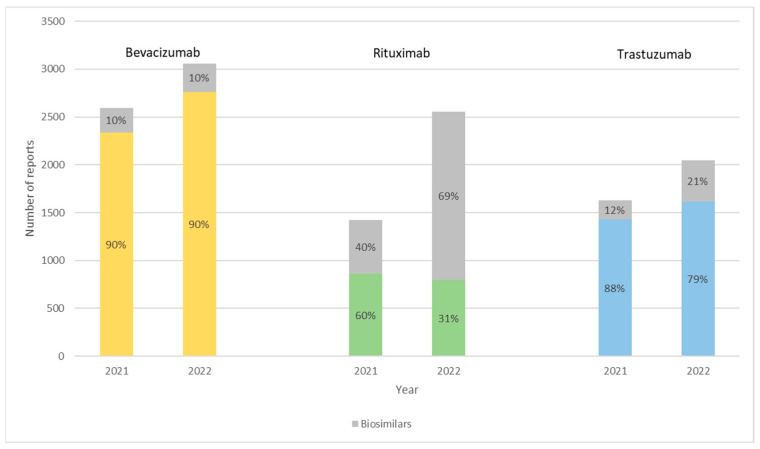
Number of reports for bevacizumab, rituximab, and trastuzumab biologics in 2021 and 2022 divided between the originators and the respective biosimilars, expressed in percentages. Yellow for *Avastin*^®^, green for *MabThera*^®^, and blue for *Herceptin*^®^.

**Table 1 cancers-15-03680-t001:** Definition of biosimilars according to international regulatory organizations.

Regulatory Authority	Definition	Reference
The European Medicines Agency (EMA)	A biosimilar is a biological medicine highly similar to another already approved biological medicine (the “reference medicine”). Biosimilars are approved according to the same standards of pharmaceutical quality, safety and efficacy that apply to all biological medicines.	The European Medicines Agency. Biosimilar Medicines: overview [1]
Food and Drug Administration (FDA)	A biosimilar is a biologic medication that is highly similar to and has no clinically meaningful differences from an existing FDA-approved biologic, called a reference product.	Food and Drug Administration. Biosimilars [9]
The World Health Organization (WHO)	A biotherapeutic product which is similar in terms of quality, safety and efficacy to an already licensed reference biotherapeutic product.	The World Health Organization. Biosimilars [10]

**Table 2 cancers-15-03680-t002:** Bevacizumab, rituximab, and trastuzumab: reference products and biosimilar agents approved in the European Union considered in our study.

	Reference Product	Authorization Date	Patent Expiry Date	Biosimilar Agent	Approval Date
Bevacizumab	* Avastin * ^ ® ^	2005	2022	*Aybintio* ^ ® ^	19 August 2020
				* Mvasi * ^ ® ^	15 January 2018
				* Zirabev * ^ ® ^	14 February 2019
Rituximab	* MabThera * ^ ® ^	1998	2013	* Blitzima * ^ ® ^	13 July 2017
				* Rixathon * ^ ® ^	15 June 2017
				* Riximyo * ^ ® ^	15 June 2017
				* Ruxience * ^ ® ^	1 April 2020
				* Truxima * ^ ® ^	17 February 2017
Trastuzumab	* Herceptin * ^ ® ^	2000	2014	* Herzuma * ^ ® ^	8 February 2018
				* Kanjinti * ^ ® ^	16 May 2018
				* Ogivri * ^ ® ^	12 December 2018
				* Ontruzant * ^ ® ^	15 November 2017
				* Trazimera * ^ ® ^	26 July 2018
				* Zercepac * ^ ® ^	27 July 2020

**Table 3 cancers-15-03680-t003:** Demographic characteristics and reporter type for reports of rituximabs.

Characteristics		Rituximab					
		Originator Biologic *MabThera*^®^ (*n* = 1657)	Biosimilar				
			* Blitzima *^ ® ^ (*n* = 7)	*Rixathon*^®^ (*n* = 422)	*Riximyo*^®^ (*n* = 39)	* Ruxience *^ ® ^ (*n* = 1024)	* Truxima *^ ® ^ (*n* = 830)
Sex	Female	872 (52.6)	1 (14.2)	237 (56.2)	28 (71.8)	683 (66.7)	189 (22.8)
	Male	753 (45.4)	3 (42.8)	169 (40)	11 (28.2)	336 (32.8)	134 (16.1)
	Unknown	32 (2)	3 (42.8)	16 (3.8)	0 (0)	5 (0.5)	507 (61.1)
Age (years)	0–17	47 (2.8)	0 (0)	17 (4)	0 (0)	3 (0.3)	20 (2.4)
	18–64	768 (46.4)	3 (42.8)	203 (48.1)	19 (48.7)	542 (52.9)	222 (26.8)
	65–85	568 (34.3)	1 (14.2)	144 (34.1)	9 (23.1)	444 (43.3)	107 (12.9)
	>85	22 (1.3)	0 (0)	11 (2.6)	1 (2.6)	21 (2.1)	6 (0.7)
	Unknown	252 (15.2)	3 (42.8)	47 (11.1)	10 (25.6)	14 (1.4)	475 (57.2)
Reporter type	Healthcare Professional	1511 (91.2)	7 (100)	407 (96.4)	38 (97.4)	980 (95.7)	783 (94.3)
	Non Healthcare Professional	146 (8.8)	0 (0)	15 (3.6)	1 (2.6)	44 (4.3)	47 (5.7)

**Table 4 cancers-15-03680-t004:** Demographic characteristics and reporter type for reports of bevacizumabs.

Characteristics		Bevacizumab			
		Originator Biologic *Avastin*^ ®^ (*n* = 5100)	Biosimilar		
			*Aybintio*^®^ (*n* = 15)	* Mvasi *^ ® ^ (*n* = 403)	*Zirabev*^®^ (*n* = 138)
Sex	Female	2088 (40.9)	9 (60)	223 (55.3)	66 (47.8)
	Male	2609 (51.2)	6 (40)	167 (41.4)	69 (50)
	Unknown	403 (7.9)	0 (0)	13 (3.2)	3 (2.2)
Age (years)	0–17	27 (0.5)	0 (0)	0 (0)	1 (0.7)
	18–64	1451 (28.5)	7 (46.7)	180 (44.7)	85 (61.6)
	65–85	2343 (45.9)	8 (53.3)	162 (40.2)	42 (30.4)
	>85	264 (5.2)	0 (0)	4 (1)	3 (2.2)
	Unknown	1015 (19.9)	0(0)	57 (14.1)	7 (5.1)
Reporter type	Healthcare Professional	4573 (89.7)	15 (100)	386 (95.8)	125 (90.6)
	Non Healthcare Professional	527 (10.3)	0 (0)	17 (4.2)	13 (9.4)

**Table 5 cancers-15-03680-t005:** Demographic characteristics and reporter type for reports of trastuzumabs.

Characteristics		Trastuzumab						
		Originator Biologic *Herceptin*^®^ (*n* = 3049)	Biosimilar					
			* Herzuma *^ ® ^ (*n* = 198)	* Kanjinti *^ ® ^ (*n* = 93)	* Ogivri *^ ® ^ (*n* = 38)	* Ontruzant *^ ® ^ (*n* = 232)	* Trazimera *^ ® ^ (*n* = 59)	* Zercepac *^ ® ^ (*n* = 2)
Sex	Female	2778 (91.1)	171 (86.4)	92 (99)	36 (94.7)	227 (97.8)	53 (89.8)	2 (100)
	Male	184 (6)	3 (1.5)	0 (0)	2 (5.3)	5 (2.2)	5 (8.5)	0 (0)
	Unknown	87 (2.9)	24 (12.1)	1 (1)	0 (0)	0 (0)	1 (1.7)	0 (0)
Age (years)	0–17	1 (0)	0 (0)	0 (0)	0 (0)	0 (0)	0 (0)	0 (0)
	18–64	1815 (59.5)	127 (64.1)	45 (48.4)	15 (39.5)	148 (63.8)	32 (54.2)	2 (100)
	65–85	542 (17.8)	35 (17.7)	18 (19.4)	11 (28.9)	77 (33.2)	23 (39)	0 (0)
	>85	30 (1)	4 (2)	2 (2.2)	0 (0)	0 (0)	0 (0)	0 (0)
	Unknown	661 (21.7)	32 (16.2)	28 (30)	12 (31.6)	7 (3)	4 (6.8)	0 (0)
Reporter type	Healthcare Professional	2655 (87.1)	195 (98.5)	84 (90.3)	36 (94.7)	229 (98.7)	52 (88.1)	2 (100)
	Non Healthcare Professional	394 (12.9)	3 (1.5)	9 (9.7)	2 (5.3)	3 (1.3)	7 (11.9)	0 (0)

**Table 6 cancers-15-03680-t006:** The most frequent and statistically significant ADRs related to Rituximab biosimilars compared to *MabThera*^®^ reported in EudraVigilance.

Rituximab.					
ADR	N	N*	ROR	CI 95%_Lower	CI 95%_Upper
COVID-19	169	46	1.73	1.42	2.09
Pruritus	163	39	1.96	1.6	2.41
Throat irritation	135	13	4.9	3.55	6.74
Blood pressure fluctuation	116	2	27.36	8.55	87.51
Malaise	112	8	6.59	4.28	10.14
Nausea	112	38	1.38	1.09	1.75
Condition aggravated	108	5	10.17	5.69	18.19
Fatigue	106	21	2.37	1.78	3.15
Headache	105	19	2.59	1.93	3.48
Hypertension	102	32	1.49	1.15	1.93
Blood pressure increased	93	32	1.36	1.04	1.77
Pain	93	7	6.24	3.85	10.13
Erythema	88	21	1.96	1.45	2.66
Cough	78	17	2.15	1.53	3.01
Arthralgia	77	8	4.51	2.82	7.22
Rheumatoid arthritis	69	7	4.62	2.76	7.74
Urticaria	63	14	2.1	1.43	3.1
Immune thrombocytopenia	59	3	9.22	3.71	22.88
Weight increased	50	6	3.9	2.13	7.14
Heart rate irregular	48	1	22.48	2.38	211.89
Throat tightness	46	6	3.59	1.94	6.64
Pain in extremity	40	6	3.12	1.65	5.89
Ear pruritus	39	1	18.24	1.88	177.4
Illness	38	3	5.92	2.24	15.65
Weight decreased	36	5	3.36	1.64	6.91
Heart rate decreased	35	2	8.18	2.2	30.38
Mantle cell lymphoma	34	1	15.9	1.6	158.05
Systemic lupus erythematosus	34	2	7.95	2.13	29.65
Chest pain	33	1	15.43	1.54	154.16
Anxiety	32	3	4.98	1.83	13.55
Joint swelling	31	3	4.83	1.77	13.2
Blood pressure systolic increased	29	1	13.55	1.33	138.53
Musculoskeletal stiffness	28	3	4.36	1.56	12.14
Therapeutic product effect incomplete	27	1	12.61	1.22	130.67
Blood pressure decreased	26	5	2.43	1.12	5.28
Fall	26	4	3.03	1.26	7.28
Paraesthesia	26	2	6.07	1.55	23.77
C-reactive protein increased	25	5	2.33	1.06	5.11
Peripheral swelling	24	4	2.8	1.15	6.82
Somnolence	24	3	3.73	1.3	10.72
Feeling hot	21	3	3.27	1.11	9.64
Abdominal discomfort	19	2	4.43	1.06	18.53

ADR adverse drug reaction, N number of suspected ADRs to biosimilars, N* number of suspected ADRs to *MabThera*^®^ ROR reporting odds ratio, CI 95%_lower lower limit of the 95% confidence interval, CI 95%_upper upper limit of the 95% confidence interval.

**Table 7 cancers-15-03680-t007:** The most frequent and statistically significant ADRs related to Bevacizumab biosimilars compared to *Avastin*^®^ reported in EudraVigilance.

Bevacizumab					
ADR	N	N*	ROR	CI 95%_Lower	CI 95%_Upper
Asthenia	32	75	2.63	1.81	3.82
Neuropathy peripheral	29	73	2.44	1.65	3.61
Abdominal pain	28	44	3.92	2.58	5.94
Blood pressure increased	28	65	2.65	1.77	3.96
Fatigue	25	92	1.66	1.1	2.52
Neutropenia	24	84	1.75	1.14	2.67
Vomiting	24	86	1.71	1.12	2.61
Constipation	19	41	2.84	1.72	4.68
Drug ineffective	19	66	1.76	1.09	2.85
Headache	19	54	2.15	1.32	3.51
Dizziness	18	27	4.09	2.39	6.99
Thrombocytopenia	18	55	2	1.22	3.3
Dyspnoea	14	40	2.14	1.21	3.8
Weight decreased	14	39	2.19	1.23	3.9
Weight increased	14	10	8.58	4.17	17.65
Epistaxis	13	42	1.89	1.05	3.42
Seizure	13	18	4.42	2.3	8.5
Chills	12	13	5.65	2.76	11.58
Cough	12	24	3.06	1.6	5.85
General physical health deterioration	12	9	8.16	3.72	17.9

ADR adverse drug reaction, N number of suspected ADRs to biosimilars, N* number of suspected ADRs to *Avastin*^®^ ROR reporting odds ratio, CI 95%_lower lower limit of the 95% confidence interval, CI 95%_upper upper limit of the 95% confidence interval.

**Table 8 cancers-15-03680-t008:** The most frequent and statistically significant ADRs related to Trastuzumab biosimilars compared to *Herceptin*^®^ reported in EudraVigilance.

Trastuzumab					
ADR	N	N*	ROR	CI 95%_Lower	CI 95%_Upper
Diarrhea	78	239	1.55	1.23	1.95
Chills	68	157	2.07	1.61	2.66
Nausea	57	169	1.6	1.22	2.1
Asthenia	38	96	1.87	1.33	2.63
Infusion related reaction	38	54	3.34	2.34	4.77
Paraesthesia	34	22	7.35	4.8	11.26
Dyspnoea	32	66	2.29	1.57	3.34
Neutropenia	32	46	3.29	2.23	4.87
Anaemia	30	58	2.44	1.65	3.62
Tremor	28	26	5.1	3.26	7.98
Arthralgia	21	31	3.19	1.95	5.22
Hypersensitivity	19	36	2.48	1.5	4.12
Pruritus	19	48	1.86	1.14	3.04
Erythema	18	21	4.04	2.31	7.04
Hypertension	18	36	2.35	1.4	3.95
Myalgia	18	19	4.46	2.53	7.87
Back pain	17	28	2.86	1.65	4.93
Thrombocytopenia	16	29	2.59	1.48	4.53
Abdominal pain	12	27	2.08	1.1	3.95
Epistaxis	12	20	2.82	1.45	5.48
Oxygen saturation decreased	12	14	4.03	1.99	8.17
Tachycardia	11	15	3.44	1.67	7.09
Hypotension	9	9	4.69	1.96	11.23
Rhinorrhoea	9	7	6.03	2.37	15.37
Hypertransaminasaemia	8	2	18.77	3.52	100.17
Drug hypersensitivity	7	6	5.47	1.88	15.93
Feeling cold	7	8	4.1	1.53	11.01
Hypokalaemia	7	13	2.52	1.03	6.16
Syncope	7	6	5.47	1.88	15.93
Blood creatinine increased	6	7	4.02	1.36	11.84
Bronchospasm	4	2	9.36	1.32	66.58
Gamma-glutamyltransferase increased	4	4	4.68	1.07	20.39

ADR adverse drug reaction, N number of suspected ADRs to biosimilars, N* number of suspected ADRs to *Herceptin*^®^ ROR reporting odds ratio, CI 95%_lower lower limit of the 95% confidence interval, CI 95%_upper upper limit of the 95% confidence interval.

## Data Availability

The data that support the findings of this study are available from the corresponding author upon reasonable request.

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
