# Peer review of "Comparative Safety Profiles of Oncology Biosimilars vs. Originators in Europe: An Analysis of the EudraVigilance Database"

_cancers, 2023, doi:10.3390/cancers15143680_

Round 1

Reviewer 1 Report

Please, see attached file

Reviewer 2 Report

The authors performed a descriptive analysis and a disproportionality study of the data reported in EV, from 01/01/2021 to 31/12/2022 for originals and biosimilar mAbs used in oncology.

            The paper is well-written and correctly structured. Please find below a few comments for your consideration:

·        Authors should present the limitations of the study

·        According to EMA guidelines, disproportionality signals were defined when cases were ≥5. Did the authors take this issue into consideration?

·        Authors should explain why they chose only 2 years for analysis.
